https://doi.org/10.1038/s41467-021-21816-x | **OPEN**

# Cryo-EM structures of HIV-1 trimer bound to CD4-mimetics BNM-III-170 and M48U1 adopt a CD4-bound open conformation

Claudia A. Jette [1], Christopher O. Barnes[1], Sharon M. Kirk[2], Bruno Melillo [2], Amos B. Smith III[2] & Pamela J. Bjorkman [1✉]

Human immunodeficiency virus-1 (HIV-1), the causative agent of AIDS, impacts millions of people. Entry into target cells is mediated by the HIV-1 envelope (Env) glycoprotein interacting with host receptor CD4, which triggers conformational changes allowing binding to a coreceptor and subsequent membrane fusion. Small molecule or peptide CD4-mimetic drugs mimic CD4's Phe43 interaction with Env by inserting into the conserved Phe43 pocket on Env subunit gp120. Here, we present single-particle cryo-EM structures of CD4-mimetics BNM-III-170 and M48U1 bound to a BG505 native-like Env trimer plus the CD4-induced antibody 17b at 3.7 Å and 3.9 Å resolution, respectively. CD4-mimetic-bound BG505 exhibits canonical CD4-induced conformational changes including trimer opening, formation of the 4-stranded gp120 bridging sheet, displacement of the V1V2 loop, and formation of a compact and elongated gp41 HR1C helical bundle. We conclude that CD4-induced structural changes on both gp120 and gp41 Env subunits are induced by binding to the gp120 Phe43 pocket.

[1] Division of Biology and Biological Engineering, California Institute of Technology, Pasadena, CA, USA. [2] Department of Chemistry, University of Pennsylvania, Philadelphia, PA, USA. ✉email: bjorkman@caltech.edu

Human Immunodeficiency Virus 1 (HIV-1) is the causative agent of Acquired Immunodeficiency Syndrome (AIDS) and currently infects over 37.5 million people[1]. The entry of HIV-1 into host target cells is initiated by binding of the host receptor CD4 to the only viral protein on the surface of HIV-1, the envelope (Env) glycoprotein, a trimer of gp120-gp41 heterodimers[2]. Env binding to CD4 induces a well-characterized set of conformational changes[3–6] that expose an occluded binding site in the gp120 V3 region for a co-receptor, either CCR5 or CXCR4[7]. Upon co-receptor binding, Env undergoes further conformational changes resulting in the insertion of the gp41 fusion peptide into the target cell membrane, allowing the fusion of the viral and host membranes and entry of the HIV-1 genetic material into the target cell[2].

X-ray and cryo-EM structures of native-like soluble HIV-1 Env trimers (SOSIPs[8]) have defined a closed, prefusion state in which the V1V2 loops at the trimer apex shield the co-receptor binding site on the V3 region[9], and a CD4-bound open state in which the gp120 subunits rotate outwards from the trimer axis, the V1V2 loops are displaced to the sides of the trimer, and the V3 loops are exposed[3–6]. A key interaction for exposure of the co-receptor binding site upon CD4 binding is the insertion of CD4 residue $Phe43_{CD4}$ into a conserved, $150Å^2$ hydrophobic cavity at the junction between the gp120 inner domain, outer domain, and bridging sheet[10]. This interaction was first observed in crystal structures of monomeric gp120 cores complexed with CD4[10], which adopt a hallmark feature of CD4-bound Env trimers in the presence or absence of CD4: a 4-stranded anti-parallel β-sheet comprising β-strands β20, β21, β2, and β3[11]. By contrast, SOSIP Env trimers in the closed, prefusion state contain a mixed parallel/anti-parallel 3-stranded β-sheet comprising strands β20, β21, and β3[12]. Upon CD4 binding to an Env trimer, the loop between strands β20 and β21 is displaced, triggering changes that are propagated through the inner domain of gp120 and resulting in trimer opening, V1V2 displacement, and 4-stranded bridging sheet formation[3–6]. Identification of the importance of the gp120 Phe43 cavity for CD4 binding led to the development of cavity-interacting small molecule and peptide compounds called CD4 mimetic (CD4m) inhibitors[13–20].

Small molecule HIV-1 entry inhibitors that prevent HIV-1 trimer opening include BMS-378806 and a related family of compounds including BMS-626529, which bind orthogonally to the Phe43 opening beneath the Env β20-21 loop and extend into the base of the Phe43 cavity. Upon inhibitor binding, the Env trimer is kept closed by allosterically preventing CD4 binding by separating the bridging sheet and the inner domain of gp120[20,21].

CD4m compounds that open trimers by binding directly into the Phe43 pocket were initially identified with the discovery of NBD-556 and NBD-557, two small molecules that inhibit HIV-1 entry into cells expressing CD4 and a co-receptor, but enhance entry into cells that express a co-receptor in the absence of CD4[13]. Subsequent studies showed that premature allosteric activation of trimer opening by these small molecules could inhibit viral entry after an initial period of increased activation[22], leading to the modification of these compounds and the development of CD4m small molecule inhibitors such as BNM-III-170 that bind to the Phe43 pocket but prevent infection of cells lacking CD4[13,14]. Members of this class were also shown to induce an intermediate Env conformation that can be stabilized by gp120 inner-domain-targeting Abs, permitting Antibody-Dependent Cellular Cytotoxicity[23].

Concurrent with the development of small molecule CD4m inhibitors, peptide CD4m inhibitors were developed using scorpion toxin scyllatoxin scaffolds in which the CDR2-like loop of CD4 containing Phe43 had been grafted[24–26]. Unlike small-molecule CD4m compounds, which primarily insert directly into the Phe43 cavity with few external interactions, CD4m peptides contain a more extensive gp120 binding interface involving not only a synthetic Phe43-equivalent residue but also an equivalent to $Arg59_{CD4}$, which forms a salt bridge with the highly-conserved $Asp368_{gp120}$[10] and an exposed C-terminal β-strand that forms hydrogen bonds with the β15 strand of gp120 immediately adjacent to the Phe43 cavity opening[15]. These peptides directly compete with CD4 binding and inhibit HIV-1 infection of cells[24,25].

Recent reports of structures of CD4-bound partially-open[5] and fully-open but asymmetric Env trimers[6] demonstrated that there are different conformations of open HIV-1 Env trimers. In addition, the structure of an Env trimer bound to the CD4-binding site antibody b12 exhibited yet another open Env conformation[4]. Here we investigated the open conformation(s) of HIV-1 Env induced by two CD4m compounds: BNM-III-170, a small molecule, and M48U1, a peptide, both of which have been structurally characterized when bound to gp120 monomeric cores[14,15]. We report single-particle cryo-EM structures of complexes of these trimer-opening CD4m compounds with the BG505 SOSIP.664 trimer[8] (hereafter BG505), which provide information about potential V1V2 displacement, V3 rearrangement, and gp41 changes that cannot be assessed in structures involving gp120 monomeric cores. These structures revealed interactions of CD4m compounds with the Phe43 cavity in complexes with three CD4m and three 17b Fabs per BG505 trimer. Inter-protomer dimensions of M48U1 and BNM-III-170-bound Env closely matched those of an open, sCD4-bound Env. In addition, the CD4m-Env structures exhibited canonical features of CD4-bound open trimer for all three protomers, including a 4-stranded bridging sheet, a displaced V1V2 loop, an exposed and displaced V3 loop, and a compact arrangement of extended gp41 HR1C helices. We conclude that BNM-III-170 and M48U1 induce Env trimers to open in a similar manner as the native CD4 ligand despite fewer contacts with gp120.

## Results

**M48U1-BG505 and BNM-III-170-BG505 complexes bind 17b IgG.** 17b, a CD4-induced (CD4i) antibody that binds Env only when the gp120 V3 loop is exposed after V1V2 loop displacement characteristic of CD4-induced Env opening, has been used as a measure of trimer opening[3–6,27,28]. We first recapitulated and extended studies showing that binding of BNM-III-170 and M48U1 CD4m compounds open Env trimers[29,30] as assessed by a 17b binding assay[31]. D7324-tagged BG505 trimers[8] were immobilized on ELISA plates by binding to the JR-52 antibody as described[8] and then incubated with either buffer, BNM-III-170, M48U1, BMS-626529, or soluble CD4 (sCD4), and the binding of CD4-induced antibodies 17b and 21c, V1V2 bNAbs BG1 and PG16, and V3 bNAb 10-1074 was measured.

BG1, PG16, and 10-1074 IgGs bound to BG505 under both closed and open conditions (Supplementary Fig. 1a). As expected, BG505 did not bind 17b or 21c IgGs in the absence of sCD4 or BNM-III-170 and M48U1 inhibitors, indicating the Env trimers were well-folded. When incubated with BNM-III-170, M48U1, or sCD4, BG505 bound to 17b IgG, confirming previous results[29,30] and demonstrating the accessibility of the V3 loop in an open state (Supplementary Fig. 1b–d). Of note, binding of 17b was lower for BG505 incubated with BNM-III-170 (Supplementary Fig. 1c), suggesting that some of the BG505 Envs were in a conformation not accessible for binding to 17b. 21c IgG bound to BG505 plus sCD4 but did not bind to BG505 incubated with BNM-III-170 or M48U1 (Supplementary Fig. 1b–d), consistent with the requirement of the epitope of this antibody spanning CD4 and gp120[32]. BG505 incubated with BMS-626529 showed

little or no binding to 17b IgG, consistent with a BG505–BMS-626529 crystal structure in the closed, prefusion state[20] (Supplementary Fig. 1a).

These results demonstrated that binding of M48U1 and BNM-III-170 caused BG505 Env trimer to adopt a conformation in which the 17b binding site on V3 was exposed.

**Cryo-EM structures of BNM-III-170-BG505-17b and BG505-M48U1-17b complexes show densities for CD4m compounds.** Although the BNM-III-170 and M48U1 CD4m compounds demonstrated 17b binding consistent with trimer opening, it was not known if other CD4-induced conformational changes in Env took place since the CD4 binding site on gp120 encompasses more than the CD4 Phe43 sidechain interacting with the gp120 Phe43 pocket (Fig. 2c). For example, other conserved interactions with gp120 include CD4 residues 29, 33, 34, 44, and 59[10].

We used single-particle cryo-EM to determine the structural effects of binding BNM-III-170 and M48U1 to an HIV-1 Env trimer. For structure determinations, BG505 was incubated with a CD4m and 17b Fab and then purified by size exclusion chromatography (SEC) to obtain CD4m-BG505-17b complexes. Samples were frozen on grids in vitrified ice and micrographs were collected on a Titan Krios microscope. 3D reconstructions were produced by iterative 2D classification, 3D classification, and 3D refinement followed by polishing[33,34]. Final reconstructions were produced for each complex at 3.7 Å for BNM-III-170-BG505-17b and 3.9 Å for M48U1-BG505-17b, as determined by the gold-standard FSC[35] (Fig. 1a, b; Supplementary Figs. 2, 3). Both structures were solved by fitting three copies of the gp120 and gp41 coordinates for open conformation A from a single-particle cryo-EM structure of sCD4-E51-BG505[6] and three copies of 17b Fab variable domain coordinates[36]. Initial models were refined without placement of CD4m compounds. Following refinement, a density that could not be accounted for by Env or 17b Fab was present within the Phe43 pockets in all gp120 protomers of both maps. Overlaying of the BNM-III-170-gp120 and M48U1-gp120 crystal structures[14,15] allowed placement of the CD4m compounds into these densities within the Phe43 pocket.

**CD4m-bound BG505 trimers displayed conformational heterogeneity.** During the processing of both CD4m-BG505 data sets, it became clear that one protomer in each structure had consistently worse density for the gp120 and 17b regions (Supplementary Fig. 4a). To determine if the lower resolution of this region in the BNM-III-170-BG505-17b complex resulted from the sub-stoichiometric binding of 17b Fab, we performed iterative rounds of 3D classification. Rather than yielding classes with different 17b binding stoichiometries, the analysis produced nearly identical classes with 17b Fab densities for all protomers and similar numbers of particles in each class regardless of the number of subclasses ($k = 4$, or 8) defined. Overlaying and alignment of the reconstructions showed that the 17b Fab with the weakest density was rotated at varying degrees away from the central axis of the Env in each subclass, but the resulting 3D classes were of poorer resolution (~6–8 Å) and precluded detailed analysis to identify differences in the conformation of the trimers in each subclass (Supplementary Fig. 2d). To improve the resolutions of the subclasses, we collected a second data set for the BNM-III-170-BG505-17b complex and repeated the analysis with more particles. Classification and analysis of the merged data produced similar 3D classes as in the first data set, but at a higher resolution (~6–7.4 Å) with close overlays of two protomers (defined as protomers 1 and 2) and different positions for protomer 3 (Supplementary Figs. 2d, 4c-d). Re-fitting gp120 and

gp41 coordinates into the gp120 densities revealed that the position of the gp120 core changed between classes and hinged as a rigid body about the gp120 β4 and β26 strands (Supplementary Fig. 4c). Since protomers in all 3D classes showed similar conformations and the overall resolution was better for the combined reconstruction, we performed analyses on the models built and refined them into the maps containing all particles without discarding classes in the final 3D classification (Fig. 1). This resulted in a 3.7 Å map of the BNM-III-170-BG505-17b complex with two well-defined gp120-gp41-17b protomers and one protomer with weaker density for 17b and gp120. As classification results for M48U1-BG505-17b were similar (Supplementary Figs. 3, 5), we also retained all particles without 3D classification for the final reconstruction at 3.9 Å resolution.

**BNM-III-170 and M48U1 bind in Phe43 pockets of BG505 Env trimer.** While the overall resolution of the BNM-III-170-BG505-17b complex was 3.7 Å, the local resolution for the gp120 Phe43 pocket was ~3.5 Å (Supplementary Fig. 4a), and with the exception of Glu478$_{gp120}$, there was density for sidechains of gp120 residues lining the Phe43 pocket in protomers 1 and 2 (Supplementary Fig. 6a). Alignment of the gp120s from the BNM-III-170-BG505-17b complex with the monomeric core gp120 from the BNM-III-170-gp120 crystal structure[14] demonstrated structural similarity (root mean square deviation, RMSD = 1.2–1.3 Å for 320 Cα atoms), and the BNM-III-170 from the gp120 core complex structure[14] aligned with the unaccounted density in the cryo-EM reconstruction (Supplementary Fig. 7a). In this position, BNM-III-170 fits into the gp120 Phe43 pocket beside the gp120 β20-β21 loop. In sCD4-bound gp120 structures, there is an 8 Å gap between the tip of the phenyl ring of the Phe43$_{CD4}$ residue and the base of the Phe43 hydrophobic cavity in gp120, leading to the development of CD4m such as BNM-III-170 that reach further into the pocket[13]. As also found for the BNM-III-170 compound in the BNM-III-170-gp120 core structure[14], the BNM-III-170 molecule bound to each protomer in the BNM-III-170-BG505-17b structure extended to the base of the Phe43 cavity (Fig. 2a).

Interactions between BNM-III-170 and gp120 in the gp120 core structure[14] occur near the entrance of the Phe43 cavity and involve H-bonds between the guanidinium of BNM-III-170 and backbone carbonyls of Arg429$_{gp120}$ and Met426$_{gp120}$ and the methylamine of BNM-III-170 with the carbonyl of Gly473$_{gp120}$. In addition, a fourth hydrogen bond is formed halfway into the Phe43 cavity between the backbone carbonyl of Asn425$_{gp120}$ and a hydrogen on the more buried nitrogen of the oxalamide linker of BNM-III-170[14]. The positioning of BNM-III-170 in the highest resolution protomer (protomer 1) of the BNM-III-170-BG505-17b structure placed its oxalamide linker within hydrogen-bonding distance of Asn425$_{gp120}$ and Gly473$_{gp120}$ (Fig. 2b), consistent with previously-reported interactions[14]. Poor density for the guanidinium of BNM-III-170 made modeling of its orientation with respect to the β-turn of the gp120 β20-β21 hairpin loop difficult. However, the density supported its placement in close proximity to backbone carbonyls of both Met426$_{gp120}$ and Asn429$_{gp120}$ (Fig. 2b), suggesting that these interactions also occur in the BNM-III-170-BG505-17b complex.

The density for M48U1 in the M48U1-BG505-17b complex was well ordered, allowing placement of its α-helix and two-stranded β-sheet into density along with the coordinates for gp120, gp41, and 17b (Fig. 1b, c). As also found for the BNM-III-170-BG505-17b complex, one protomer of the BG505 trimer showed weaker density for 17b, gp120, and M48U1 (Supplementary Fig. 7b).

The cyclohexylmethoxy phenylalanine side chain at M48U1 position 23 occupies a structurally-equivalent position with

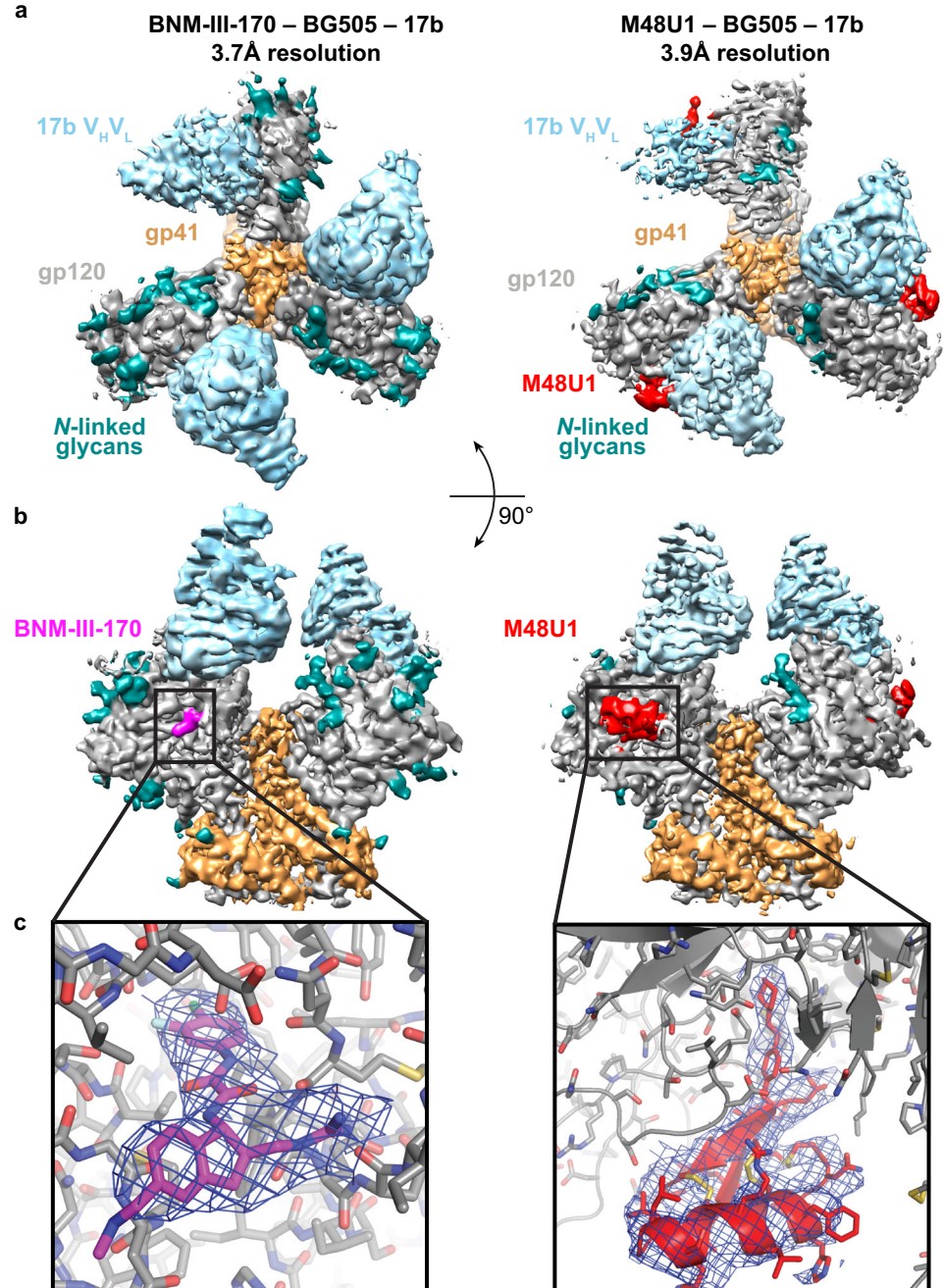

**Fig. 1 Cryo-EM structures of BNM-III-170-BG505-17b and M48U1-BG505-17b. a** Top-down view of density maps for BNM-III-170-BG505-17b and M48U1-BG505-17b complexes. **b** Side view of density maps for BNM-III-170-BG505-17b and M48U1-BG505-17b complexes. Boxed region indicates binding site for one CD4m molecule on each structure. **c** Close-up views of densities (blue) in CD4m binding sites. Densities are shown at 7σ. Structure colors: 17b variable domain = light blue, gp120 = gray, gp41 = light orange, N-linked glycans = teal, BNM-III-170 = magenta, M48U1 = red, carbon atoms = gray, magenta, or red, oxygen atoms = red, nitrogen atoms = blue, sulfur atoms = yellow, chlorine atoms = green, fluorine atoms = pale blue.

respect to $Phe43_{CD4}$. Whereas $Phe43_{CD4}$ inserts only 8 Å into the gp120 cavity, the hydrophobic cyclohexylmethoxy phenylalanine inserts and extends ~11.5 Å from its Cα, reaching the base of the gp120 Phe43 cavity (Fig. 2a). Unlike BNM-III-170, all polar contacts between M48U1 and gp120 occur outside of the Phe43 cavity. The β−strand spanning residues $Cys24_{M48U1}$ to $Cys26_{M48U1}$ (equivalent to $Leu44_{CD4}$ to $Lys46_{CD4}$) forms hydrogen bonds with backbone atoms of residues $Asp368_{gp120}$, $Gly367_{gp120}$, and $Gly366_{gp120}$ (Fig. 2b). In previous crystal structures, $Asp368_{gp120}$ was identified as an important binding residue both for $Arg59_{CD4}$ and for the M48U1-equivalent residue

$Arg9_{M48U1}$[10,15]. However, reduced sidechain density for the M48U1 helix in the M48U1-BG505 structure limited the accurate placement of sidechains.

**BNM-III-170 and M48U1 open BG505 trimer to a similar degree as CD4.** To evaluate conformations of HIV-1 Env, we previously used distance measurements between equivalent residues within gp120 subunits of an Env trimer, from which we could compare the degree of gp120 opening between trimers in closed, b12-bound, and sCD4-bound states[5,6,37]. Here we used

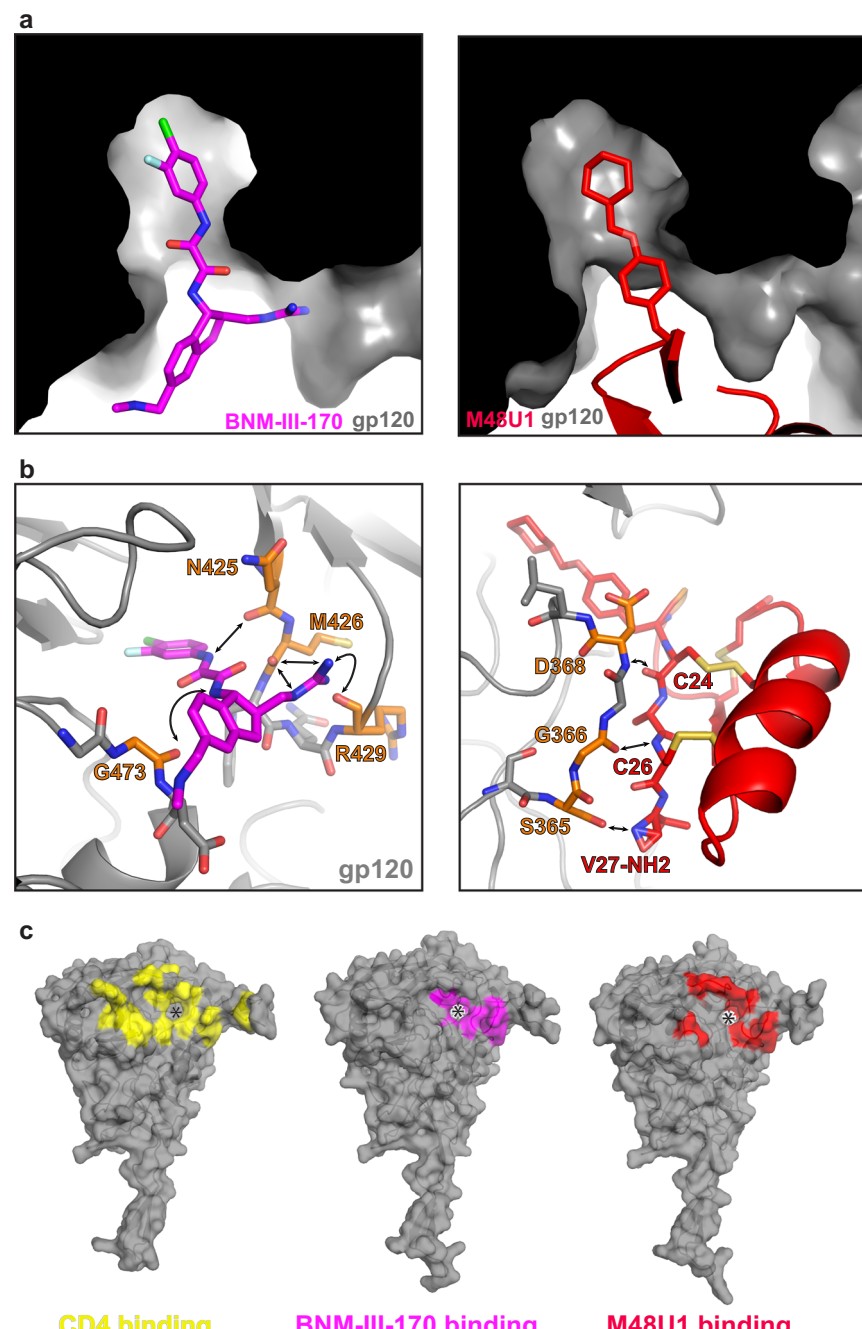

**Fig. 2 BNM-III-170 and M48U1 binding to the gp120 Phe43 pocket.** Atom colors: carbon = magenta (BNM-III-170), red (M48U1), or orange (gp120), nitrogen = blue, oxygen = red, sulfur = yellow, chlorine = green; fluorine = cyan. **a** Cut-away side view of gp120 showing BNM-III-170 or M48U1 inserting into Phe43 pocket cavity of gp120 (black/gray). **b** Left: Stick model of BNM-III-170 within gp120 Phe43 pocket. Potential interactions between BNM-III-170 and backbone atoms of gp120 residues indicated by an arrow pointing to colored atoms of gp120 residues (sidechains omitted for clarity). Right: Stick and cartoon model of M48U1 within gp120 Phe43 pocket. **c** Surface rendering of gp120 showing interacting residues for sCD4 (PDB: 6U0L [10.2210/pdb6U0L/pdb], left, yellow), BNM-III-170 (middle, magenta), or M48U1 (right, red). Highlighted residues are 4 Å or less from the bound molecule. gp120 = gray, * = Phe43 cavity entrance.

this method to assess the effects of BNM-III-170 and M48U1 binding on the BG505 conformation (Fig. 3). Measurements for the CD4m-BG505 complexes were complicated by the lack of three-fold Env trimer symmetry due to the heterogeneity of one of the protomers (designated as protomer 3 in each complex) (Supplementary Figs. 4–5). Thus the measurements between equivalent residues in protomers 1 and 2 are more accurate than measurements between protomers 2 and 3 and between protomers 1 and 3. For distance measurement comparisons with

sCD4-bound Env trimers, we averaged distances from conformations A and B of an asymmetric sCD4-BG505-E51 Fab complex[6] and a symmetric sCD4-B41-17b complex[4] (Table 1). We also averaged measured distances between protomers for each CD4m-BG505 complex. We report a single distance for three-fold symmetric Env structures and an average distance with a standard deviation for asymmetric structures in order to more clearly address whether the CD4m-Env structures adopt what can be described as an sCD4-bound open Env trimer structure (Table 1).

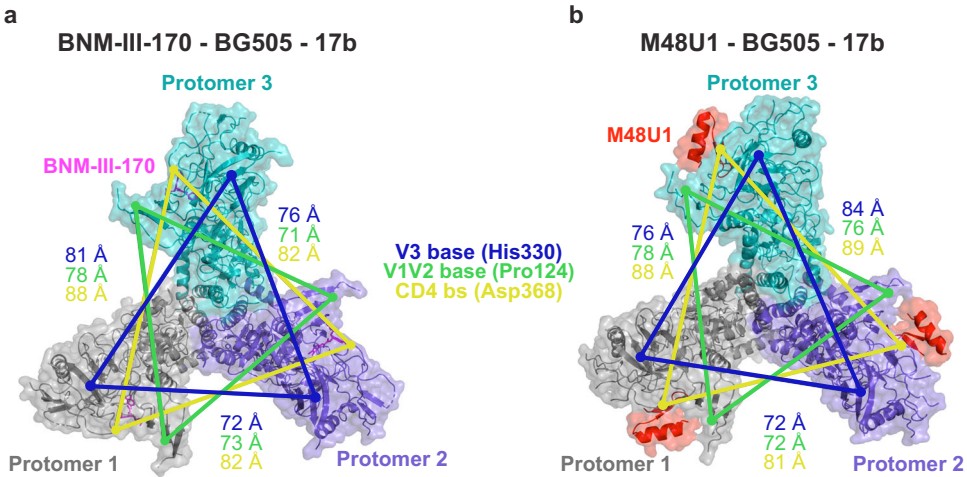

**Fig. 3 BNM-III-170-bound and M48U1-bound Env inter-protomer dimensions match those of open CD4-bound Env.** Structure colors: Protomer 1 = gray, protomer 2 = purple, protomer 3 = teal, BNM-III-170 = magenta, M48U1 = red. Top-down view of surface and cartoon rendering of **a**, BNM-III-170-bound and **b**, M48U1-bound Env trimer structures showing inter-protomer distance measurements between reference residues for the base of the V3 loop (His330$_{gp120}$, blue), the base of the V1/V2 loop (Pro124$_{gp120}$, green), and the CD4 binding site (CD4bs, Asp368$_{gp120}$, yellow). 17b Fabs have been removed for clarity.

**Table 1 Interprotomer distances of envelope trimers.**

**Average distances (Å)\***

| Structure | PBD code | State | V3 Base (His330$_{gp120}$) | V1/V2 Base (Pro124$_{gp120}$) | CD4bs (Asp368$_{gp120}$) |
|---|---|---|---|---|---|
| BG505-IOMA-10-1074 | 5T3Z | Closed | 69 | 14 | 54 |
| BMS-626529-BG505-PGT122-35O22 | 5U70 | Closed | 69 | 14 | 55 |
| b12-B41 | 5VN8 | Open | 75 ± 0.3 (n = 3) | 69 ± 0.2 (n = 3) | 85 ± 0.0 (n = 3) |
| sCD4-Env-17b | 5VN3, 6U0L, 6U0N | Open | 74 ± 4.0 (n = 9) | 77 ± 5.9 (n = 9) | 80 ± 5.0 (n = 9) |
| BNM-III-170-BG505-17b | This study | Open | 76 ± 4.6 (n = 3) | 74 ± 3.5 (n = 3) | 84 ± 3.4 (n = 3) |
| M48U1-BG505-17b | This study | Open | 77 ± 5.9 (n = 3) | 75 ± 2.8 (n = 3) | 86 ± 4.2 (n = 3) |

\*For asymmetrically open structures, distances shown are mean ± s.d. of n number of protomers.

As previously described, the V3 regions of closed Env and b12-bound open Env are occluded by the V1V2 loop[5] (Fig. 4a, b). Opening of b12- or sCD4-bound Env involves rotation of the gp120 as a rigid body away from the central gp41 helices, hinging on the loops connecting the β26 and β4 strands to the gp120 core.[5,38] A hallmark of sCD4, but not b12, binding to Env trimers is the displacement of V1V2 to expose the coreceptor binding site on V3 and the resulting disorder of most of the V1V2 and V3 loops[3,5,6]. These conformational changes have corresponding changes in the positioning of residues in the V1V2 loop, the V3 loop, and the CD4 binding site (CD4bs) that can be evaluated by measuring between the three copies of Pro124$_{gp120}$ at the V1V2 base, the three copies of His330$_{gp120}$ at the V3 base, and the three copies of Asp368$_{gp120}$ at the CD4bs. A typical closed Env structure[39] displayed V1V2 distances of 14 Å and V3 distances of 69 Å (Table 1). Similarly, an Env trimer that was kept in a closed conformation by the Phe43 cavity-binding small molecule BMS-626529[20] showed V1V2 and V3 inter-protomer distances of 14 Å and 55 Å, respectively. In sCD4-liganded open Env, the displacement of V1V2 from the trimer apex to the sides of the Env trimer resulted in inter-protomer V1V2 distances of 77 Å ± 5.9 Å and V3 distances of 74 Å ± 4 Å.

The BNM-III-170-BG505-17b and M48U1-BG505-17b structures both showed similar inter-protomer measurements as sCD4-bound Envs for V1V2 displacement (74 Å ± 3.5 Å and 75 Å ± 2.8 Å, for the BNM-III-170 and M48U1 complexes, respectively) and V3 positioning (76 Å ± 4.6 Å and 77 Å ± 5.9 Å, respectively). In addition, as found in CD4-bound open

structures[3–6], most of the V1V2 and V3 loops were disordered in the CD4m-bound Env structures.

Opening of both b12- and CD4-bound trimers leads to hinging about the loops connecting the β26 and β4 strands to the main portion of the gp120 subunit and rotation of the gp120 as a rigid body away from the central gp41 helices.[5,38] This is reflected in changes of the average inter-protomer distances between Asp368$_{gp120}$ residues in the CD4 binding site: from 54 Å and 55 Å in closed Env structures to 80 Å ± 5.0 Å in CD4-bound open Env and 85 Å ± 0 Å for b12-bound open Env. The analogous measurements for the CD4m-BG505 complexes (84 Å ± 3.4 Å and 86 Å ± 4.2 Å) suggested that CD4m binding induced equivalent gp120 rotation and displacement indicative of trimer opening.

Taken together, the inter-protomer distances for V1V2, V3, and the CD4 binding site provide quantitative verification that BNM-III-170 and M48U1 induce an open BG505 structure similar to the sCD4-bound open conformation.

**BNM-III-170 and M48U1 induce additional structural changes similar to those induced by sCD4 binding.** In addition to gp120 rotation and displacement to create an open Env trimer, sCD4-bound Env structures exhibit structural changes within the gp120 and gp41 subunits compared with closed Env structures. In order to determine if the CD4m-bound Env structures demonstrated similar conformational changes as sCD4-bound open Envs, we compared specific regions of closed and open Env

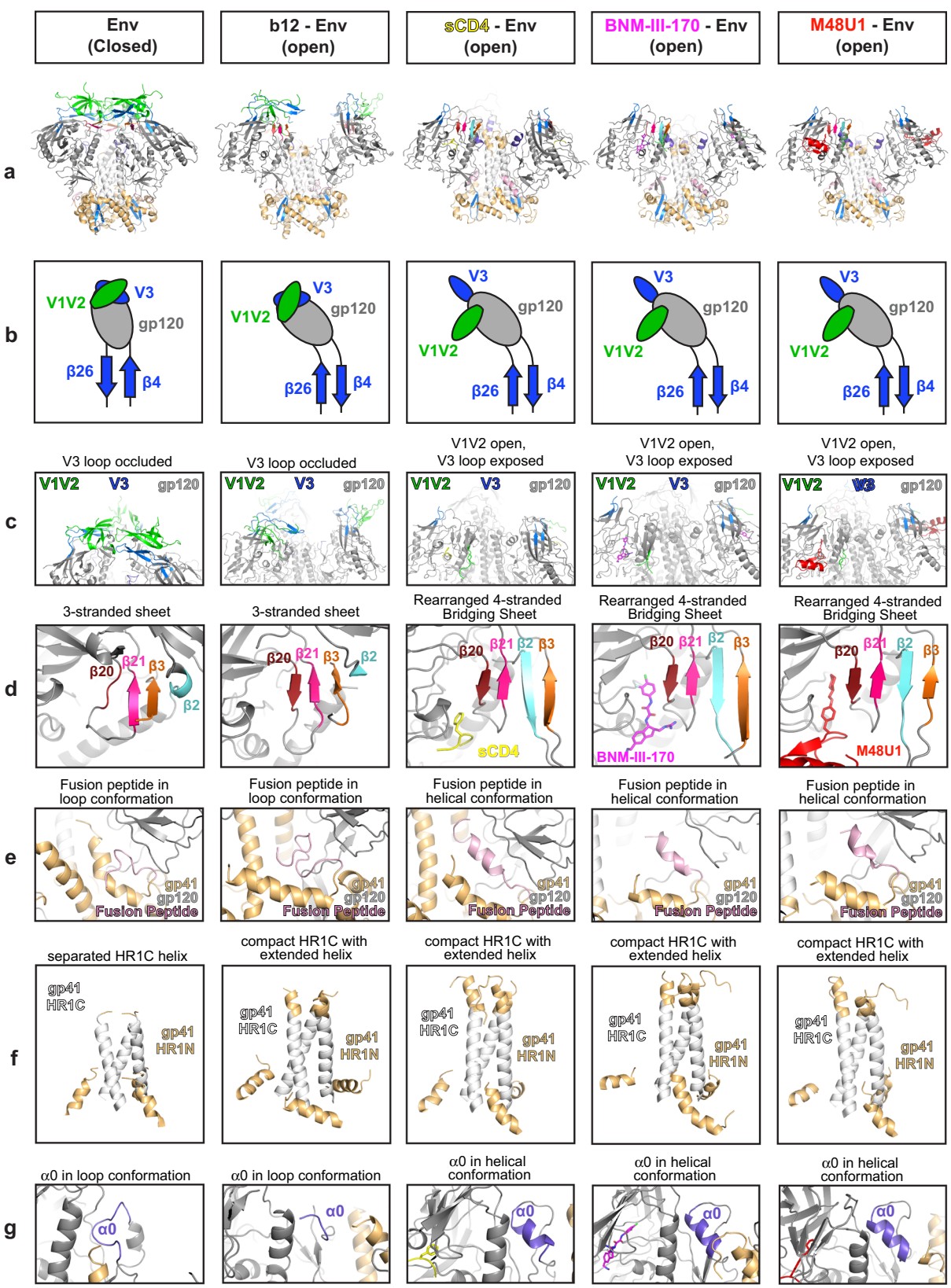

structures. For comparisons with sCD4-bound Env trimer, we choose conformation A from a structure of sCD4-BG505-E51 Fab[6] that differs from a slightly different conformation (conformation B) also observed for the asymmetric sCD4-BG505-E51 complex[6] and for a symmetric sCD4-B41-17b complex[4]. We chose conformation A for comparisons because, like our CD4m-

bound open Env structures, the fusion peptide was ordered in conformation A (Fig. 4e), but was disordered in conformation B.

A large conformational change that occurs upon sCD4 binding to Env trimer is the displacement of V1V2 to expose the coreceptor binding site on V3[3,5,6]. As previously described, the V3 regions of closed Env and b12-bound open Env are occluded

**Fig. 4 Conformational features of gp120 and gp41 in structures of closed and open Envs.** Cartoon and schematic models showing features of the HIV-1 Env trimers in the closed conformation (PDB 5T3Z [10.2210/pdb5T3Z/pdb]), b12-bound open conformation (PDB 5VN8 [10.2210/pdb5VN8/pdb]), sCD4-bound open conformation (PDB 6U0L [10.2210/pdb6U0L/pdb], Conformation A), the BNM-III-170-bound open conformation, and the M48U1-bound open conformation. Structure colors: gp120 = gray, gp41 = light orange, CD4 Phe43 loop = yellow, BNM-III-170 = magenta, M48U1 = red, V1V2 loop = green, V3 loop = blue, β20 strand = dark red, β21 strand = hot pink, β3 strand = orange, β2 strand = cyan, HR1C helix = white, fusion peptide = light pink, α0 loop = purple. **a** Cartoon depiction of BG505 Env with regions of interest colored. **b** Schematic of gp120 angle with relation to the β26/β4 β-strands and V1V2 and V3 loop positioning. **c** V1V2 and V3 loop positions. **d** 3-stranded β-sheet (β20, β21, β3 β-strands) versus 4-stranded bridging sheet (β20, β21, β2, β3 β-strands). **e** Fusion peptide conformation. **f** gp41 HR1C helix conformation (gp120 N-terminal portion of gp41 removed for clarity). **g** α0 loop versus α0 helix conformation.

by the V1V2 loop[5] (Fig. 4a–c). The CD4m-bound open Env complexes showed displacement of the V1V2 loop and exposure of the V3 loop in a similar manner as in sCD4-bound open Env (Fig. 4b–c).

Accompanying the opening of the V1V2 loop, the 3-stranded β-sheet formed by the β20, β21, and β3 strands in closed Env structures[12] undergoes a rearrangement upon binding of CD4 in which β2 becomes an ordered β-strand and swaps positions with the β3 strand, forming a 4-stranded β-sheet[5,6,38] called the bridging sheet[10] (Fig. 4d). Although the b12-bound Env structure can be classified as open with respect to its gp120 positions[38], it retains the 3-stranded β-sheet found in closed Env structures (Fig. 4d), likely because the V1V2 and V3 regions move as a rigid body with gp120 rather than V1V2 being displaced to the sides of Env[5]. In common with sCD4-bound open Env structures, CD4m-bound Envs included 4-stranded bridging sheets (Fig. 4d).

The fusion peptide also exists in several conformations: an unstructured loop in closed and b12-bound open structures versus a helical conformation in sCD4-bound and CD4m-bound open conformations (Fig. 4e). In addition, the rotation and repositioning of the gp120 subunits upon trimer opening permits the rearrangement of the gp41 helices to form a compact HR1C helical bundle, as also found in b12-bound and sCD4-bound open Env trimers (Fig. 4f). The CD4m-BG505 complexes adopted the same gp120 positioning and gp41 rearrangements as found in sCD4-bound and b12-bound Envs (Fig. 4f). While the HR1C helix became more compact upon rearrangement in the CD4m-bound structures, it also extended and formed several additional ordered helical turns at the tip of the gp41 bundle that makes up part of HR1N (Fig. 4f), as also found in the b12-bound and sCD4-bound open structures, therefore its occurrence in CD4m-bound open Env structures suggests this is a conformational change that typically occurs upon trimer opening.

In closed or b12-bound Envs, the gp120 α0 region nestled against the top of the gp41 helices is in an unstructured loop (Fig. 4g). When sCD4 is bound, the α0 adopts a helical structure and is located at the top of the HR1 helix of the adjacent protomer (Fig. 4g). Likewise, the CD4m-BG505 open structures showed analogous placement and helical α0 conformations to the sCD4-bound structure for the three protomers in each Env trimer (Fig. 4g). We conclude that the CD4m-bound Envs exhibit structural changes within the gp120 and gp41 subunits characteristic of sCD4-bound open Env structures.

## Discussion

Viral fusion protein flexibility is required for their functions in fusing the viral and host cell membranes[2]. Indeed, HIV-1 Env trimers exhibit different degrees of opening in response to external signals[9]. Here, we investigated how the activating CD4m molecules BNM-III-170 and M48U1 alter the conformation of Env trimers. Since the CD4m-gp120 interface is smaller than the sCD4-gp120 interface, it was possible that rather than adopting a fully open conformation normally induced by host receptor binding, activating CD4m molecules could induce a partially-

open conformation (e.g., similar to sCD4 plus 8ANC195-bound Env trimers[5]) or an open conformation without V1V2 displacement as in the b12-bound Env trimer[4]. Alternatively, since CD4m molecules have little to no bulk that could interact outside the gp120 Phe43 pocket, they could also allow the trimer to adopt a previously-unseen open conformation due to limiting steric clashes that would occur in the presence of bound CD4.

Using single-particle cryo-EM, we found that two CD4m compounds, BNM-III-170 and M48U1, bound to the native-like BG505 Env trimer resulted in open trimer structures similar to sCD4-bound structures, both in terms of inter-subunit gp120 rotation and displacement and in terms of intra-subunit conformational changes. These results demonstrate that interactions of small molecule compounds at the gp120 Phe43 pocket are sufficient to cause Env trimer opening and structural rearrangements similar to those induced by the CD4 host receptor. These results can be used to inform the design of CD4m compounds as possible therapeutics.

## Methods

**Protein expression and purification.** A construct encoding the BG505 SOSIP.664 native-like envelope gp140 trimer including stabilizing mutations (A501C_{gp120}, T605C_{gp120}, I559P_{gp41}), an introduced glycosylation site (T332N_{gp120}), an improved furin protease cleavage site (REKR to RRRRRR), and truncation after residue 664 in gp41[8] was subcloned into the pTT5 expression vector (National Research Council of Canada) and transiently expressed in HEK293F cells. BG505 trimer was purified from the supernatant by 2G12 Fab immunoaffinity chromatography followed by SEC using a Superdex 200 Increase 10/300 GL column (GE Life Sciences) running in TBS (20 mM Tris pH 8, 150 mM NaCl) plus 0.02% NaN₃ as described[37]. BG505 trimer that was C-terminally tagged with the D7324 sequence[31] was prepared in the same way. For some experiments, BG505 SOSIP.664 was expressed and purified from supernatants of a stable CHO cell line (kind gift of John Moore, Weill Cornell Medical College) as described[40].

Expression plasmids encoding JR-52 IgG were the kind gift of James Robinson (Tulane University) and John Moore (Weill Cornell Medical College). Expression plasmids encoding the heavy and light chains of 17b, BG1, 21c, PG16, 10-1074, JR-52 IgGs were transiently co-transfected into Expi293F cells (Gibco) using Expofectamine (Invitrogen). IgGs were purified from supernatants by protein A chromatography (GE Life Sciences) followed by SEC using a Superdex 200 Increase 10/300 GL column (GE Life Sciences). IgGs were stored in TBS. 6x-His-tagged version of 17b Fab and sCD4 (domains 1 and 2 of CD4; amino acids 1-186) were expressed as described[5].

**CD4 mimetic compounds.** The (+)(R,R)BNM-III-170 small molecule (referred to as BNM-III-170 throughout the manuscript) was synthesized as described[14,41] and stored at −20°C in DMSO until use. Lyophilized M48U1 peptide (sequence reported in ref. [15]) was purchased from Presto Pepscan Inc. (Lelystad, The Netherlands) and resuspended in DMSO before use.

**ELISA.** 96-well plates (Corning, #9018) were coated with JR-52 IgG at 5 μg/mL in 0.1 M NaHCO₃ pH 8.6 at 4 °C overnight. sCD4-BG505, BNM-III-170-BG505, M48U1-BG505, and BMS-626529 complexes were prepared by incubating CD4m with D7324-tagged BG505 trimer[31] at a 15:1 small molecule to trimer ratio or a 6:1 sCD4 to trimer overnight at room temperature in TBS. Plates were blocked on the following day for 1 h with TBS-TMS (20 mM Tris pH 8, 150 mM NaCl, 0.05% Tween 20, 1% non-fat dry milk, 1% goat serum (Gibco 16210-072)). Complexes diluted to a final concentration of 10 μg/mL in TBS-TMS were incubated on coated plates for 1 h at room temperature and three 10-minute washes were performed using TBS-T (20 mM Tris pH 8, 150 mM NaCl, 0.05% Tween 20). IgG versions of 17b, 21c, BG1, PG16, and 10-1074 were diluted from 20 μg/μL to 1 ng/μL in 2-fold increments. Plates with trimer complexes were incubated with IgGs for 2 h at room

temperature, followed by 3 washes of TBS-T, and then incubated for 30 mins at room temperature with anti-human IgG HRP at 1:4000 (Southern Biotech #2040-05). 5 washes of TBST were done followed by development using 1-Step Ultra TMB-ELISA Substrate Solution (ThermoFisher Scientific, 43028) and quenching with 1 N HCl. Quantification of results was performed using a plate reader detecting absorbance at 450 nm. All samples were evaluated in duplicate ($n = 2$). After averaging duplicates, individual data points were graphed and figures were made using Graphpad Prism v8.

**Cryo-EM sample preparation.** BNM-III-170-BG505-17b and M48U1-BG505-17b complexes were assembled by incubating CD4m compounds BNM-III-170 or M48U1 with BG505 overnight at room temperature at a molar ratio of 10:1 (CD4m:trimer). 17b Fab was added the next day at a 9:1 ratio (Fab:trimer) and incubated at room temperature for 2–4 h. Complexes were purified by SEC on a Superdex 200 Increase GL 50/150 or a Superdex 200 Increase 10/300 GL column (GE Healthcare) and fractions containing CD4m-BG505-17b complexes were concentrated to 1.4-1.5 mg/mL. Cryo-EM grids were frozen using a Mark IV Vitrobot (ThermoFisher) at 22 ºC and 100% humidity. 3.1 μL of the sample was applied to Quantifoil R2/2 300 mesh grids, blotted for 3 or 3.5 s, and plunge frozen into liquid ethane. Grids were then transferred to grid boxes in liquid nitrogen and stored until data collection.

**Cryo-EM data collection and processing.** Cryopreserved grids were loaded into a Titan Krios electron microscope (ThermoFisher) equipped with a GIF Quantum energy filter (slit width 20 eV) operating at 300 kV and a nominal 80,000 magnification. Images were recorded using a K3 direct electron detector (Gatan) in counting mode with a pixel size of 1.104 Å/pixel and defocus range of 1–3.5 μm using SerialEM acquisition software (ver. 3.8 Beta). Images were exposed for a total dose of 40 or 60e⁻/pixel fractionated into 40 subframes. Micrographs were manually curated after motion correction with MotionCor2[42] and the contrast transfer function was fit with Gctf ver. 1.06[43] to remove cracked or icy micrographs. Initial particles from 100 randomly selected micrographs were picked using the RELION autopicker[33,34], and reference-free two-dimensional (2D) classes were generated in RELION ver. 3.0. Particles from good initial classes were used to generate Ab initio models in CryoSPARC ver. 2.15.0[44]. RELION autopicker was then used to pick particles from all micrographs and subjected to 2D reference-free classification in RELION. Good 2D classes were selected and subjected to two rounds of 2D classification. The 3D classification was performed on 2D averages using RELION[33,34]. Rounds of 3D classification were attempted with different numbers of subclasses ($k = 4, 8$) for data sets. The final 3D classification used for analysis included 4 subclasses ($k = 4$) and 1 round of classification for M48U1-BG505-17b and 4 subclasses ($k = 4$) and 2 iterative rounds for merged data sets of BNM-III-170-BG505-17b. The 3D glasses were used for analysis; however, no 3D classes were discarded for the final round of 3D classification before proceeding to 3D refinement and polishing because higher resolution maps for the areas of interest (Env and CD4m compounds) were obtained when using all particles. 3D reconstructions were produced in RELION 3D auto-refine using ab initio models as starting models[33,34]. CTF correction and polishing were performed in RELION, and final maps were generated after a final round of 3D auto-refining. Sphericity was calculated using the 3DFSC server[45].

**Model building.** Coordinates of gp120 (PDB 6U0L [10.2210/pdb6U0L/pdb], Conformation B), gp41 (PDB 6U0L [10.2210/pdb6U0L/pdb], Conformation B), and 17b Fab $V_H$-$V_L$ domains (PDB 2NXY [10.2210/pdb2NXY/pdb] were fitted into map density using UCSF Chimera ver. 1.14[46]. Coordinates were initially refined using phenix.real_space_refine[47] from the Phenix package ver. 1.18.1[48] and manually refined using Coot v0.8.9.1[49]. Initial refinement rounds were performed without placing CD4m compounds. Placement of BNM-III-170 or M48U1 was done in UCSF Chimera by overlaying the refined gp120 portions of our cryo-EM structures with corresponding X-ray crystal structures of M48U1-gp120 (PDB: 4JZZ [10.2210/pdb4JZZ/pdb]) or BNM-III-170-gp120 (PDB: 5F4P [10.2210/pdb5F4P/pdb]), which placed CD4m compounds into unambiguous, unaccounted-for density within the Phe43 pocket region of gp120. CD4m were then rigid body fit in Chimera to better fit the density. Further rounds of manual and automated refinement of models containing CD4m were done. As the resolution was not sufficient to determine conformations of M48U1 BNM-III-170 in the third protomer, the conformations from the crystal structures were modeled into the density using rigid body fitting and were not further refined. In addition, we trimmed side chains to Cβ of M48U1-BG505-17b protomer 3 gp120,17b $V_H$-$V_L$ and M48U1 (except for the cyclohexylmethoxy phenylalanine at position 23) due to poor resolution. Coordinates from final models were rigid-body fit into 3D subclasses using phenix.real_space_refine[48] followed by rigid body fitting for the β4/β26 strands in Coot[49].

**Structural analysis.** Structure figures were made using UCSF Chimera[46] or PyMOL[50]. Unless otherwise noted, figures showing a single gp120-gp41 protomer were made using one of the two protomers (protomers 1 and 2) in each complex showing the best density. Potential hydrogen bonds were assigned as interactions that were <4.0 Å and with A-D-H angle >90°. Potential van der Waals interactions

between atoms were assigned as interactions that were <4.0 Å. Due to low resolution, hydrogen-bond and van der Waals interaction assignments should be considered tentative. Interacting residues for Fig. 2c were determined as residues within 4 Å of bound CD4m or sCD4 using PyMOL[50]. Inter-protomer Cα distances were measured between Cα atoms using the Measurement Wizard tool in PyMOL version 2.3.2. Average interprotomer distances in Table 1 were calculated as the mean between distances of each protomer □ the standard deviation. Distances for three-fold symmetric structures are listed without a standard deviation. Distances for sCD4-bound open Env structures were derived from two asymmetric structures of sCD4-BG505-E51 (PDBs 6U0L [https://doi.org/10.2210/pdb6U0L/pdb] and 6U0N [https://doi.org/10.2210/pdb6U0N/pdb]) and one structure of a more symmetric sCD4-B41-17b (PDB 5VN3 [https://doi.org/10.2210/pdb5VN3/pdb]).

Pairwise Cα alignments between CD4-bound gp120-core structures and CD4m-bound Env gp120 core structures in Supplementary Fig. 8 were done using the alignment function PyMOL v2.3.2 without excluding outliers. Atoms belonging to regions that were not present in both gp120 and gp120 core structures were excluded.

**Reporting summary.** Further information on research design is available in the Nature Research Reporting Summary linked to this article.

## Data availability

The structural coordinates were deposited into the Worldwide Protein Data Bank (wwPDB) with accession codes 7LO6 [https://doi.org/10.2210/pdb7LO6/pdb] (BNM-III-170-BG505-17b) and 7LOK [https://doi.org/10.2210/pdb7LOK/pdb] (M48U1-BG505-17b). EM density maps were deposited into EMDB with accession numbers EMD-23462 (BNM-III-170-BG505-17b) and EMD-23465 (M48U1-BG505-17b). Other data are available upon reasonable request.

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

## Acknowledgements

Cryo-EM was performed in the Beckman Institute Resource Center for Transmission Electron Microscopy at Caltech with assistance from directors A. Malyutin and S. Chen. We thank J. Vielmetter and the Beckman Institute Protein Expression Center at Caltech for protein production, J.E. Robinson (Tulane University) for the JR-52 antibody, John Moore (Weill Cornell Medical College) for the BG505 stable cell line. This work was supported by the Bill and Melinda Gates Foundation Collaboration for AIDS Vaccine Discovery (CAVD) grant INV-002143 (P.J.B.) and the National Institute of Allergy and Infectious Diseases (NIAID) Grant numbers 2 P50 AI150464 and HIVRAD P01 AI100148 (P.J.B.) and AI50471 and GM56550 (A.B.S.). A portion of this research was supported by NIH grant U24GM129547 and performed at the PNCC at OHSU and accessed through EMSL (grid.436923.9), a DOE Office of Science User Facility sponsored by the Office of Biological and Environmental Research.

## Author contributions

C.A.J. designed experiments, purified proteins, assembled protein, and cryo-EM samples, collected cryo-EM data, processed cryo-EM data, performed model building, and refinement, analyzed data, performed ELISA experiments, and wrote the manuscript. C.O.B. designed experiments, purified proteins, collected cryo-EM data, assisted in data processing, assisted in model building and refinement, and assisted in ELISA experiments. S.M.K. and B.M. developed and synthesized BNM-III-170. A.B.S. supervised and guided BNM-III-170 development. P.J.B. supervised and guided the project and wrote the manuscript.

## Competing interests

The authors declare no competing interests.
