## [Peer Review File · Nature Communications]

Reviewers' Comments:

Reviewer #1:

Remarks to the Author:

Jette et al. present cryo-EM structural data for the HIV-1 BG505 SOSIP Trimer bound to either BNM-III-170 or M48U1 and the CD4i antibody 17B. Both compounds/constructs are known to induce trimer opening from its prefusion closed state. The authors first demonstrate this using an ELISA assay followed by investigation of the structure of these complexes by cryo-EM. The primary conclusion from this work is that the open state previously observed in a CD4 bound open state trimer is observed in the presence of these ligands despite their having fewer contacts with gp120. Although of general interest to the field, the novelty of the investigation could be significantly improved with a more in depth examination of the conformational heterogeneity reported in the manuscript. Suggestions are as follows:

The authors indicate that maps were refined from particles without a 3D classification step. Is there a particular reason 3D classification as a particle set "cleaning" step was not performed? There is potential for misfolded particles, ice, etc. to make it through 2D classification. The authors show results from 3D classifications of the particles with differing trimer states. Does this include all classes from the classification? If not, a 3D refinement excluding particles assigned to any structurally ambiguous classes should be performed and reported.

The conformational heterogeneity analysis is of great interest. Considering the size of the BNM-III-170 dataset, the authors have an opportunity to quantify the degree of mobility in the open state gp120. This could be accomplished using symmetry expansion and masking of a single protomer. Though resolutions of the classes obtained may limit residue level fits, rigid body fitting of gp120 and gp41 would likely be possible for measurements. The motion may be of a continuous nature as suggested by the fact that the presented classification yields similar particle counts for each class. A bound, however precise, for this flexibility would nevertheless be of interest.

The authors mention the hinge motion of gp120 relative to gp41 yet measure residue distances between protomers. Though these distances report movement of gp120 a direct measure would be better for the comparisons with previously published structures. This should include the change in hinge angle and any rotation of gp120 about its long axis. Such measures would be particularly useful for the above suggestion to quantify the degree of gp120 open state conformational variability.

Examination of the internal conformational changes quantitatively via a method such as distance difference matrix (DDM) approach would be helpful to aid in comparison with previous structures.

Minor Comments:

Page 3 line 117: A space appears to be missing at "bind17b".

Page 3 line 138: Suggest a supplemental figure displaying a structure highlighting sites of interaction of the CD4m compounds and CD4.

Page 5 line 207-208: The sentence beginning with "Poor density" is incomplete.

Page 5 line 239-242: Why average distances for the two conformations? The authors should explain the rationale for averaging rather than comparing the actual values.

Page 7 line 333: Misspelling; "confirmations" should be "conformations"

Reviewer #2:

Remarks to the Author:

This paper reports on the cryoEM structures of a soluble mimic of HIV viral spike BG505 bound to CD4 mimetics M48U1 and BNM-III-170 and 17b, a CD4-induced antibody. Additionally the reports used additional biophysical methods to confirm the results observed by cryoEM.

Their conclusion is that the CD4 mimetics induce the same conformational changes as the CD4 receptor, despite fewer contacts on gp120.

The studies are well conducted, with no apparent flaws in the data analysis, interpretation and conclusions. However the results are not unexpected and it is unclear how this work will be relevant to the field. These molecules have been named CD4 mimetic because they mimic CD4 and induce the same conformational changes in gp120, despite fewer contacts on gp120 as the authors state. This study confirms this while adding information on gp41 structural conformational changes, which is of interest although it is unclear how this will relate to the functional spike since likely gp120 will dissociate from gp41 once the CD4m bind.

Can the authors indicate an advantage of using these CD4m over sCD4, were they able to determine structures of regions that were not resolved before. It is not clear while reading the text. For example, are the V3 and V1V2 regions resolved? They don't appear to be - which will be good to mention.

Can the authors describe in more details the conformational heterogeneity? What do they think is happening? How does it compare with the asymmetric state they reported before?

When doing the structural comparison with sCD4-bound Env trimer, the authors state: "For comparisons with sCD4-bound Env trimer, we choose conformation A from a structure of sCD4-BG505-E51 Fab6 286 that differs from a slightly different conformation (conformation B) also observed for the asymmetric sCD4-BG505-E51 complex6 287 and for a symmetric sCD4-B41-17b complex4 288 ." Can they explain why they chose that conformation and not the others (it will remind the readers what is different between these other conformations (that have been published)).

fig 2b - can the authors use the same orientation for each CD4m or add panels to that effect. Additionally, it could be useful to show overlays of the compounds in the BG505/17b cryoEM structures vs gp120 cores.

fig 3. The authors claim that the distance match the one from the CD4-BG505 complex but the ones from the b12-BG505 complex could also match, maybe the authors can explain in more details why they think the structures with the CD4m are closer to the CD4-BG505 than the b12-BG505 ?

Minor changes

-line 119 - M48U1

-Supl Fig1 legend - a and b are switched

Would suggest to use a quaternary specific antibody such as PGT145 to confirm the "closed" state of the trimer.

-line 149 - what is open conformation A - maybe the authors can add details

-lines 196-197: sentence needs to be rewritten

As also found in the BNM-III-170- gp120 core structure14, the BNM-III-170 molecules extended to the base of each Phe43 cavity of each protomer in the BNM-III-170-BG505-17b structure (Fig. 2a).

-lines 208-209- words appear to be missing

Poor density for the guanidinium of BNM-III-170 made modeling of its orientation with respect to the β -turn of the gp120 the β 20- β 21 hairpin loop.

Reviewer #1 (Remarks to the Author):

Jette et al. present cryo-EM structural data for the HIV-1 BG505 SOSIP Trimer bound to either BNM-III-170 or M48U1 and the CD4i antibody 17B. Both compounds/constructs are known to induce trimer opening from its prefusion closed state. The authors first demonstrate this using an ELISA assay followed by investigation of the structure of these complexes by cryo-EM. The primary conclusion from this work is that the open state previously observed in a CD4 bound open state trimer is observed in the presence of these ligands despite their having fewer contacts with gp120. Although of general interest to the field, the novelty of the investigation could be significantly improved with a more in depth examination of the conformational heterogeneity reported in the manuscript. Suggestions are as follows:

We thank the reviewer for these suggestions for how to add novelty to our paper.

The authors indicate that maps were refined from particles without a 3D classification step. Is there a particular reason 3D classification as a particle set "cleaning" step was not performed? There is potential for misfolded particles, ice, etc. to make it through 2D classification. The authors show results from 3D classifications of the particles with differing trimer states. Does this include all classes from the classification? If not, a 3D refinement excluding particles assigned to any structurally ambiguous classes should be performed and reported.

Thank you for pointing this out. As explained in the revised Methods we made our reconstructions using all of the particles that came from the final 2D classification step without a 3D classification step for removing junk particles.

Following iterative rounds of 2D classification, we found that there were no junk classes containing bad particles in the 3D classifications in the M48U1-BG505-17b. We did not change our analysis of the M48U1-BG505-17b map or particle set as it did not appear to have any junk particles from our classification analysis.

Following 3D classification, the combined BNM-III-170-BG505-17b data did have a small 3D class containing what appears to be junk particles, which we did not show in our manuscript (Supplementary Fig. 4b). We have added this junk class to our figure for clarity (Supplementary Fig. 2b). In order to remove these junk particles, we pooled the particles of the first three classes and performed a second round of 3D classification. This yielded 4 classes very similar to those for the M48U1-BG505-17b classification with no junk class. We took the BNM-III-170-BG505-17b particles from the second round of 3D classification and performed an additional round of 3D refinement and postprocessing. The map from the postprocessing using the 'junk-free' particle set was at the same resolution as our previously-reported reconstruction and showed no differences in features. Our new reconstruction fit our previously-refined model reasonably well, and we performed one additional round of refinement to yield a new model that better fit the new reconstruction. Although there were no major differences between our previously-reported map and the new map or our previously-reported model and our new model, we updated all the figure panels showing the BNM-III-170-BG505-17b structures to show the new reconstruction and model that contained no junk classes.

With the exception of the junk class in the BNM-III-170-BG505-17b classification that was not shown in the original manuscript, all other classes for our 3D classification are shown in our manuscript. Consequently, there are no ambiguous classes in our data. As described in the main text, the differences present between each subclass for our structures can be attributed to the positioning of the third protomer's gp120 containing a bound 17b Fab, which is at varying

counter-clockwise rotated positions. Similar to protomer 3 though to a lesser extent, there are also some differences among the subclasses in positioning of the gp120 and 17b Fab of Protomer 2. Differences in Protomer 2 gp120 and 17b positioning are much less apparent in the overall reconstruction containing all the particles. The structural elements from each subclass were analyzed by rigid body fitting gp120 and gp41 subunits from the final refined model. All subclasses fit each gp120 and gp41 well and had unaccounted-for density inside all of the Phe43 pockets that could fit the respective CD4m molecules present in the complex.

We experimented with classifying our data sets into varying numbers of 3D classes (k=4 or 8). Classifying the first BNM-III-170-BG505-17b data set into 8 classes yielded one subclass that looked like it may have been missing a 17b Fab in the complex. Further sub-classification of this class into 4 more classes yielded 4 identical classes, all with density for 3 Fabs bound to the trimer. We therefore concluded that differences in the classes could be attributed to positioning of the 17b/gp120 of protomers rather than changes in other structural elements. Consequently, we chose not to further refine and analyze the individual subclass maps as they were of lower resolution than the map containing all the particles. Instead, we focused on the map containing all particles for each structure.

The argument could be made that the 17b Fab density for each protomer is slightly worse in some subclasses. We included 17b Fab in our complex in order to better align our particles, but the 17b structure and interface has already been well characterized (Ozorowski et al. 2017; Wang/Barnes et al. 2018) and was not a focus of this study. Since including all classes in the final refinement yielded a better map for our regions of interest (Env and CD4m), we did not discard any classes based on 17b density.

The conformational heterogeneity analysis is of great interest. Considering the size of the BNM-III-170 dataset, the authors have an opportunity to quantify the degree of mobility in the open state gp120. This could be accomplished using symmetry expansion and masking of a single protomer. Though resolutions of the classes obtained may limit residue level fits, rigid body fitting of gp120 and gp41 would likely be possible for measurements. The motion may of a continuous nature as suggested by the fact that the presented classification yields similar particle counts for each class. A bound, however precise, for this flexibility would nevertheless be of interest.

Thank you for this suggestion. While there are cases where particle expansion and masking has worked well for further analyzing single particle EM structures, we have not found this to be helpful in previous open Env structures our lab has published (e.g., Wang/Barnes et al. 2018) and did not use it here. We did try masking of one or two protomers without particle expansion and this yielded a lower resolution map that had less features in the areas of interest.

We chose to rigid body fit the gp120 subunit directly into the 3D classes for each data set. We have added additional panels Supplementary Fig. 4c to better show the comparison of the gp120 subunit and the extent to which it is different in each class for BNM-III-170-BG505-17b.

The authors mention the hinge motion of gp120 relative to gp41 yet measure residue distances between protomers. Though these distances report movement of gp120 a direct measure would be better for the comparisons with previously published structures. This should include the change in hinge angle and any rotation of gp120 about its long axis. Such measures would be particularly useful for the above suggestion to quantify the degree of gp120 open state conformational variability.

We have experimented with various ways of quantifying the degree of trimer openness over the years, settling on inter-protomer distance measurements as the most informative. We have used inter-protomer distance measurements for defining closed, partially-open, and various open states of HIV-1 Env trimers in papers since 2015 (e.g., (1-5)), therefore we believe this method is best for comparisons with previous literature. It also allows assessment of potential asymmetries (e.g., Yang et al., 2019 below). Regarding the suggestion by the reviewer: the position of the long axis of gp120 would vary depending on the position of the V1V2 loops in that gp120 (ordered and “on top” of gp120 in closed trimers and open b12-bound trimers versus disordered and to the “side” on CD4-bound open trimers), thus this type of measurement would be complicated to interpret.

1. R. P. Galimidi *et al.*, Intra-spike crosslinking overcomes antibody evasion by HIV-1. *Cell* **160**, 433–446 (2015).
2. L. Scharf *et al.*, Broadly Neutralizing Antibody 8ANC195 Recognizes Closed and Open States of HIV-1 Env. *Cell* **162**, 1379-1390 (2015).
3. H. Wang *et al.*, Cryo-EM structure of a CD4-bound open HIV-1 envelope trimer reveals structural rearrangements of the gp120 V1V2 loop. *Proc Natl Acad Sci U S A* **113**, E7151-E7158 (2016).
4. H. Wang, C. O. Barnes, Z. Yang, M. C. Nussenzweig, P. J. Bjorkman, Partially Open HIV-1 Envelope Structures Exhibit Conformational Changes Relevant for Coreceptor Binding and Fusion. *Cell Host Microbe* **24**, 579-592 e574 (2018).
5. Z. Yang, H. Wang, A. Z. Liu, H. B. Gristick, P. J. Bjorkman, Asymmetric opening of HIV-1 Env bound to CD4 and a coreceptor-mimicking antibody. *Nat Struct Mol Biol* **26**, 1167-1175 (2019).

Examination of the internal conformational changes quantitatively via a method such as distance difference matrix (DDM) approach would be helpful to aid in comparison with previous structures.

Thank you for this suggestion. We looked into the DDM approach (e.g., as in Huang et al. 2005) and agree that this is a useful thing to do when comparing gp120 core structures with and without CD4m compounds. The problem in comparing our structures to previous CD4m bound gp120 structures is that our structures are at considerably lower resolutions. For example, the gp120 structures compared in Huang et. al. were at 2.2Å, 2.75Å, and 2.9Å, whereas our structures are at 3.7Å and 3.9Å. As the differences in Huang et al. were subtle (the RMSD values reported for overall distance differences were as small as 0.159Å), we would not be able to conclude if differences we see between our structures and the previous structures are a consequence of resolution differences or actual conformational changes.

In addition, the most useful DDM comparisons to perform with our structures would be to published gp120/CD4m crystal structures containing BNM-III-170 (PDB 5F4P) or M48U1 (PDB 4JZZ). We feel that distance comparisons such as these would not be fair because we used these structures as starting models to build our structures and, particularly in regions where the density of the reconstruction is poorly defined, it was necessary to rely heavily on the starting models to guide the refinement.

Minor Comments:

Page 3 line 117: A space appears to be missing at "bind17b".

Thank you for finding this typo, which has been corrected.

Page 3 line 138: Suggest a supplemental figure displaying a structure highlighting sites of

interaction of the CD4m compounds and CD4.

Thank you for this suggestion. We have added panels c to Fig. 2 comparing CD4 points of interaction with the gp120 with the BNM-III-170 and M48U1 points of interaction.

Page 5 line 207-208: The sentence beginning with "Poor density" is incomplete.

Thank you for finding this typo, which has been corrected.

Page 5 line 239-242: Why average distances for the two conformations? The authors should explain the rationale for averaging rather than comparing the actual values.

Thank you for bringing this issue to our attention. As explained in the revised manuscript (end of first paragraph in the "BNM-III-170 and M48U1 open BG505 trimer to a similar degree as CD4" section, we wanted to address a general question: i.e., are the CD4m-bound trimers in a conformation similar to an average open HIV-1 Env trimer? If we reported all possible distances (e.g., between all combinations of protomers in asymmetric structures, between protomers in symmetric structures), we thought the table would become confusing and fail to make our main point. The degree of potential asymmetry in open conformations is reflected in larger standard deviations for the measurements, which we believe is informative and succinctly makes one of the points we were trying to make. Since the coordinates and density maps will be available, these measurements could be made by an interested reader.

Page 7 line 333: Misspelling; "confirmations" should be "conformations"

Thank you for finding this typo, which has been corrected.

Reviewer #2 (Remarks to the Author):

This paper reports on the cryoEM structures of a soluble mimic of HIV viral spike BG505 bound to CD4 mimetics M48U1 and BNM-III-170 and 17b, a CD4-induced antibody. Additionally the reports used additional biophysical methods to confirm the results observed by cryoEM. Their conclusion is that the CD4 mimetics induce the same conformational changes as the CD4 receptor, despite fewer contacts on gp120.

The studies are well conducted, with no apparent flaws in the data analysis, interpretation and conclusions. However the results are not unexpected and it is unclear how this work will be relevant to the field. These molecules have been named CD4 mimetic because they mimic CD4 and induce the same conformational changes in gp120, despite fewer contacts on gp120 as the authors state.

These compounds were called CD4 mimetics because they mimic the interaction of CD4 residue Phe43 in the Phe43 pocket of gp120, not because the nature of their potential induced conformational changes to Env trimers were known. Prior to our structures, there was no way of knowing if the compounds would also induce changes induced by CD4 binding, such as V1V2 displacement, formation of the 4-stranded gp120 bridging sheet (note that the b12 antibody that recognizes the CD4 binding site does NOT induce these changes despite opening Env trimer), formation of compact gp41 helices, and the other changes listed in Fig. 4.

This study confirms this while adding information on gp41 structural conformational changes, which is of interest although it is unclear how this will relate to the functional spike since likely gp120 will dissociate from gp41 once the CD4m bind.

Changes in the gp41 helices is of interest because a compact HR1C (as seen in CD4-bound and CD4m-bound Env trimers) was found in the post-fusion 6-helical bundle structures of gp41 (as discussed in Wang/Barnes et al., 2018; cited above and in our paper). Thus CD4 binding, and as we now know, also CD4m binding, induces allosteric changes on the way to the gp41 conformation involved in membrane fusion. We find it very interesting that not only does CD4 binding induce these long-range changes, but so also do small molecule compounds. Thus CD4m compounds mimic CD4 to an extent that was previously unknown.

Can the authors indicate an advantage of using these CD4m over sCD4, were they able to determine structures of regions that were not resolved before. It is not clear while reading the text. For example, are the V3 and V1V2 regions resolved? They don't appear to be - which will be good to mention.

Thank you for pointing out that the text did not explain that most of the V1V2 and V3 regions are not resolved in these structures or in previous CD4-bound structures. This is now discussed in the revised text and shown in Fig. 4. Regarding potential advantages of CD4m over sCD4 for therapies or in vivo experiments, we note that sCD4, but not CD4m compounds, could have off-target effects due to binding class II MHC molecules.

Can the authors describe in more details the conformational heterogeneity? What do they think is happening? How does it compare with the asymmetric state they reported before?

Previously reported asymmetric open CD4-bound Env in Yang et al. 2019 showed protomers with different conformations as well as differences in inter-protomer distances. All three protomers were ordered equally and therefore we could describe these asymmetric features.

In our structures reported here, we found that one protomer was less ordered than the others. Analysis of the overall reconstructions showed differences in the inter-protomer distances in our structures (Fig. 3) that generally matched other open structures. As explained in the main text, there is some uncertainty in these measurements since our particles had a mixed population of gp120 positions. Through extensive 3D classification, we were able to conclude that this was because the position of the 17b-bound gp120 subunit of Protomer 3 (and to a lesser extent Protomer 2) was different between 3D subclasses. In spite of differences in positioning of the gp120 subunits, the subclasses and overall reconstruction of our CD4m-BG505-17b open structures clearly show similar structural conformations for all three protomers (Fig. 4). These conformations match one of the conformations of the asymmetrically open CD4-bound Env reported previously (Conformation A, Fig. 1c for reviewers below).

When doing the structural comparison with sCD4-bound Env trimer, the authors state: "For comparisons with sCD4-bound Env trimer, we choose conformation A from a structure of sCD4-BG505-E51 Fab6 286 that differs from a slightly different conformation (conformation B) also observed for the asymmetric sCD4-BG505-E51 complex6 287 and for a symmetric sCD4-B41-17b complex4 288 ." Can they explain why they chose that conformation and not the others (it will remind the readers what is different between these other conformations (that have been published)).

We chose conformation A because the structural elements in the open conformation A protomer matched our CD4m-bound open trimer protomers (See Fig. 4 in our paper). This has been clarified by the addition of this sentence in the main text: "We chose conformation A for

comparisons because, like our CD4m-bound open Env structures, the fusion peptide was ordered in conformation A (Fig. 4e) but was disordered in conformation B.”

Please see Fig. 1 for reviewers below for more details about differences between conformation A and conformation B in the previous asymmetric sCD4-bound Env trimer structures (Fig. 6c,d from Yang et al. 2019).

Fig. 1: Comparison of open conformations A and B in Figure 6 of Yang et al. 2019. **c**, Open conformation A showing a rearranged 4-stranded Bridging sheet, a helical α_0 , a helical fusion peptide, and an extended, compacted HR1C helix. **d**, Open conformation B showing a rearranged 4-stranded Bridging sheet, a helical α_0 , a fusion peptide in the loop conformation, and an extended, compacted HR1c helix.

fig 2b - can the authors use the same orientation for each CD4m or add panels to that effect Additionally, it could be useful to show overlays of the compounds in the BG505/17b cryoEM structures vs gp120 cores.

In Figure 2, we chose orientations for the bound CD4m panels to best show all of the interacting regions of the CD4m and gp120. To clarify the relative binding orientations, we have added the CD4m compounds for each protomer to the alignments in Supplementary Fig. 7, which are in the same orientation for both the BNM-III-170 and M48U1-bound structures.

fig 3. The authors claim that the distance match the one from the CD4-BG505 complex but the ones from the b12-BG505 complex could also match, maybe the authors can explain in more details why they think the structures with the CD4m are closer to the CD4-BG505 than the b12-BG505 ?

As shown in Fig. 4, CD4 and CD4m binding, but not b12 binding, induce displacement of the V1V2 loops and exposure of V3.

*Minor changes
-line 119 - M48U1*

Thank you for finding this typo, which has been corrected.

-Supl Fig1 legend - a and b are switched

Thank you for finding this mistake, which has been corrected.

Would suggest to use a quaternary specific antibody such as PGT145 to confirm the “closed” state of the trimer.

We and others have solved at least 50 X-ray and cryo-EM structures of the BG505 trimer, without ever seeing anything other than a closed state (unless b12, sCD4, or a CD4m compound is bound).

-line 149 - what is open conformation A - maybe the authors can add details

Fig. 4b summarizes the relevant structural details of the sCD4-bound conformation A, which differs slightly from sCD4-bound conformation B, as described in Yang et al., 2019. See Fig. 1 for reviewers above for details.

-lines 196-197: sentence needs to be rewritten

As also found in the BNM-III-170- gp120 core structure 14, the BNM-III-170 molecules extended to the base of each Phe43 cavity of each protomer in the BNM-III-170-BG505-17b structure (Fig. 2a).

Done.

-lines 208-209- words appear to be missing

Poor density for the guanidinium of BNM-III-170 made modeling of its orientation with respect to the β -turn of the gp120 the β 20- β 21 hairpin loop.

Thank you for finding this problematic sentence, which has been revised.

We thank the reviewers for their careful reading of our paper and hope that the revised version is now suitable for publication in Nature Communications.

Sincerely,
Pamela J. Bjorkman

Reviewers' Comments:

Reviewer #1:

Remarks to the Author:

The authors answers are sufficient.

Reviewer #2:

Remarks to the Author:

The authors have addressed some comments but not all raised by the reviewers.